# Identification of Axinellamines A and B as Anti-Tubercular Agents

**DOI:** 10.3390/md22070298

**Published:** 2024-06-28

**Authors:** Emily J. Strong, Lendl Tan, Sasha Hayes, Hayden Whyte, Rohan A. Davis, Nicholas P. West

**Affiliations:** 1School of Chemistry and Molecular Biosciences, and the Australian Infectious Diseases Research Centre, The University of Queensland, Brisbane, QLD 4072, Australia; 2Griffith Institute for Drug Discovery, School of Environment and Science, Griffith University, Brisbane, QLD 4111, Australia; 3NatureBank, Griffith University, Brisbane, QLD 4111, Australia

**Keywords:** natural products, NatureBank, tuberculosis, *Mycobacterium*, alkaloid, axinellamine, haplosclerida, marine sponge, biodiscovery

## Abstract

Tuberculosis remains a significant global health pandemic. There is an urgent need for new anti-tubercular agents to combat the rising incidence of drug resistance and to offer effective and additive therapeutic options. High-throughput screening of a subset of the NatureBank marine fraction library (*n* = 2000) identified a sample derived from an Australian marine sponge belonging to the order Haplosclerida that displayed promising anti-mycobacterial activity. Bioassay-guided fractionation of the organic extract from this Haplosclerida sponge led to the purification of previously identified antimicrobial pyrrole alkaloids, axinellamines A (**1**) and B (**2**). The axinellamine compounds were found to have a 90% minimum inhibitory concentration (MIC_90_) of 18 µM and 15 µM, respectively. The removal of protein and complex carbon sources reduced the MIC_90_ of **1** and **2** to 0.6 and 0.8 µM, respectively. The axinellamines were not toxic to mammalian cells at 25 µM and significantly reduced the intracellular bacterial load by >5-fold. These data demonstrate that axinellamines A and B are effective anti-tubercular agents and promising targets for future medicinal chemistry efforts.

## 1. Introduction

*Mycobacterium tuberculosis* is the primary causative agent of tuberculosis (TB). TB is known as the leading cause of death from a single infectious agent [1]. Treatment regimens for *M. tuberculosis* infections are intensive and extensive [2], a fact further exacerbated by the increasing threat of multi-drug-resistant (MDR) and extensively drug-resistant (XDR) TB [3]. The journey to eliminate TB is a long one, and the discovery of new therapeutics is essential in the on-going fight against TB.

Since the discovery of the first antibiotic, natural products have traditionally been the undisputed major source of antimicrobial agents [4]. While target-based drug discovery has constituted the bulk of drug development efforts over the past decades, there has been little success in that regard towards the development of anti-TB therapies [5]. In contrast, renewed interest in tapping into the vast resource of natural products to screen for novel compounds has yielded encouraging results [6,7,8]. Marine-derived products alone have seen the successful identification of hundreds of compounds that possess anti-TB therapeutic effects [9]. Considering the vast diversity of marine natural products, the full extent of this valuable resource remains to be explored.

In this study, we describe the anti-TB screening of a subset of the NatureBank marine-derived fraction library (*n* = 2000, from a total of 105,000 fractions) [10,11]. High-throughput screening (HTS) utilising the fast-growing mycobacteria *Mycobacterium smegmatis* identified 10 prioritised fractions. Bioassay-guided fractionation of the CH_2_Cl_2_/MeOH extract from a Haplosclerida sponge led to the purification of the known antimicrobial pyrrole alkaloids axinellamine A (**1**) and axinellamine B (**2**) [12], which we determined to be responsible for the anti-tubercular activity observed during the screening of the fraction library. 

## 2. Results

### 2.1. HTS of the NatureBank Fractions and Bioassay-Guided Fractionation

We sought to identify new natural compounds that inhibit mycobacterial growth. To increase the screening capacity, the initial HTS was conducted using the fast-growing *M. smegmatis* as a surrogate for *M. tuberculosis*. Biota with fractions identified as inhibiting *M. smegmatis* were subjected to bioassay-guided fractionation, and the reprocessed fractions were tested against virulent *M. tuberculosis*. Compounds within the fractions were then purified, identified, and further assayed against *M. tuberculosis* to determine their efficacy. 

A total of 2000 fractions obtained from NatureBank [11], representing the natural chemistry of hundreds of different Australian biota, were initially screened in duplicate via separate experiments using fluorescent *M. smegmatis*. This initial HTS work resulted in the selection of 10 fractions from 10 different marine invertebrates (Table 1), all of which showed significant anti-mycobacterial activity. These 10 NatureBank fractions were followed up with large-scale extraction and bioassay-guided fractionation studies.

The freeze-dried and ground specimens were sequentially extracted with *n*-hexane, CH_2_Cl_2_:MeOH, and MeOH. The *n*-hexane extract was discarded since it contained highly lipophilic (log P > 5) material, while all CH_2_Cl_2_ and MeOH extracts were combined and dried to give a crude extract. This extract was subjected to reversed-phase HPLC as part of a standard NatureBank bioassay-guided workflow [13,14], resulting in the generation of 60 timed fractions. These fractions were screened against *M. tuberculosis* in a resazurin-based microplate assay. Of the 10 original hit biota, 8 did not contain any HPLC fractions that inhibited *M. tuberculosis* growth (Appendix A). A bioassay analysis of all 60 HPLC fractions from the Haplosclerida hit demonstrated promising *M. tuberculosis* inhibition (Figure 1a) in the material eluting between 41 and 43 min (Figure 1b). 

### 2.2. Identification of Anti-Tubercular Marine Natural Products from the Prioritised Haplosclerida Sponge

HPLC-derived fractions 41 to 43 from the Haplosclerida extract demonstrated the highest *M. tuberculosis* inhibition. Following lyophilization, these fractions (41 and 43) yielded the TFA salts of the previously described marine natural products axinellamine A (**1**) and axinellamine B (**2**), respectively, while fraction 42 was a mixture of these compounds. NMR, MS, and chiro-optical data comparisons of compounds **1** and **2** unequivocally determined these molecules to be axinellamines A and B (Figure 2) [12].

Axinellamines are imidazo−azolo−imidazole alkaloids first isolated from the Australian marine sponge *Axinella* sp. [12]. Of note, the only structural differences between axinellamines A and B relate to the configurations at C-5 and C-9 (see Figure 2). Furthermore, several syntheses of axinellamines have been reported in the literature [15,16,17]. Although only **2** was initially shown to be mildly effective as an antimicrobial agent against *Helicobacter pylori* [12] and *Bacillus subtilis* [18], a subsequent study utilising synthetics **1** and **2** reported varying minimum inhibitory concentrations across several Gram-negative and Gram-positive bacteria [16]. 

### 2.3. In Vitro Anti-Tubercular Efficacy of **1** and **2**

To determine the inhibition of **1** and **2** against *M. tuberculosis*, a minimum inhibitory concentration (MIC) resazurin reduction assay was utilised [19]. Using standard, complex growth media, **1** had an MIC_90_ of 18 µM and **2** of 15 µM (Figure 3a). Compounds **1** and **2** have previously been reported as having protein-binding potential, reducing their potency [16]. The complex growth media traditionally used in mycobacterial MIC assays contain a high level of bovine serum albumin and tryptone (0.5% and 1% final concentration, respectively). Due to this and the confounding effects it may cause, we repeated MIC assays in minimal media with and without protein supplementation (10% foetal bovine serum). In minimal media without protein supplementation, **1** and **2** had MIC_90_ values of 0.6 and 0.8 µM, respectively. Of note, the addition of foetal bovine serum only had minimal effects on MIC, with an increase in the MIC_90_ of **1** and **2** to 1.2 and 1.4 µM, respectively (Figure 3b).

*M. tuberculosis* is an intracellular pathogen that resides within macrophages during infection. We therefore next wanted to assess the cytotoxicity of the compounds on an *M. tuberculosis* infection-relevant cell type prior to assessing intracellular bacterial killing efficacy. For cytotoxicity evaluation, an MTT assay was conducted [20] utilising RAW264.7 macrophages. No toxicity was observed at concentrations up to 25 µM of **1** and **2**, with or without protein supplementation (Figure 4a). To determine the intracellular efficacy of axinellamines, RAW264.7 macrophages were infected with *M. tuberculosis* and treated with **1** or **2**. After 24 h of treatment, the intra-macrophage load of *M. tuberculosis* was enumerated. A significant 10-fold reduction in intracellular *M. tuberculosis* was demonstrated after just 24 h of treatment with **1** or **2** (Figure 4b).

## 3. Discussion

We screened 2000 natural fractions from the NatureBank fraction library [10] and identified 10 fractions from seven marine sponges, one sea anemone, one sea squirt, and one bryozoan that inhibited *M. smegmatis* growth (Table 1). This screening success, using the model organism, *M. smegmatis*, led to further testing against *M. tuberculosis*. Unfortunately, eight of the initially prioritised hit biota failed to demonstrate any inhibition against *M. tuberculosis* after follow-up large-scale extraction and bioassay-guided fractionation work (Appendix A). While the original biomaterial from these fractions did inhibit mycobacteria growth, this discrepancy is likely accounted for in the different bacterial strains. Although *M. smegmatis* allows for convenience in HTS, it is used as a rapidly growing, avirulent model for *M. tuberculosis*. One of the strengths of this screening system is that it allows for broad identification of material that can then be retested in the relevant pathogenic organism. It was surprising that *M. smegmatis* inhibition did not correlate to *M. tuberculosis* inhibition in more instances. It is generally accepted that the most important consideration when using *M. smegmatis* as a surrogate in drug screening is sensitivity. Many compounds display less potency against *M. smegmatis* than they do for *M. tuberculosis* [21], but not so in our experience with this collection. However, one marine-derived fraction from Haplosclerida did exhibit strong anti-tubercular activity. 

The processed Haplosclerida extract had multiple UV-active peaks in the reversed-phase HPLC spectrum (Figure 1b). The most potent inhibition was observed with fractions that eluted from the column at ~40 min (fractions 41 to 43) (Figure 1a). These peaks were subsequently identified to be axinellamines A and B. However, we were unable to identify reports of anti-tubercular activity for axinellamine A or B, despite these compounds being established as antibacterial in some settings [12,16,18]. Marine natural products have previously been identified with anti-tubercular activity, with alkaloids being amongst the most represented compound classes [9,22,23]. Axinellamines are imidazo−azolo−imidazole alkaloids, which are a sub-class of pyrrole–imidazole alkaloids. In general, pyrrole-imidazole alkaloids are believed to contain antifouling properties and thus likely play a defensive ecological role for marine sponges [24]. 

In line with previous biological reports [16], axinellamines A and B have very similar MICs, with axinellamine A displaying slightly more potent activity (Figure 3). Here, we show only marginal increases in MIC in the presence of protein (Figure 3b), unlike previously described studies [16]. More significantly, the reduction in complex carbon sources, such as Tween-80 and glucose available in DMM, reduced MICs by up to 30× (Figure 3b). Available carbon has been shown to drastically alter bacterial sensitivity to antimicrobials in vitro. Both rifampicin and isoniazid, the current front-line drugs used for TB treatment, are also affected by the carbon source due to cell envelope remodelling and the cellular redox state due to carbon catabolism, respectively [25]. While further work is required to elucidate the mechanism of action for axinellamines A and B, the enhanced effectiveness observed in a simple carbon source demonstrates some insight. Potentially, the cell envelope remodelling by *M. tuberculosis* when grown in simple carbon sources could simply allow for the enhanced import of axinellamines A and B into the bacteria, or, potentially, axinellamines may inhibit components of central carbon metabolism. This may be a particularly interesting future line of enquiry, given that treatment of *E. coli* with axinellamine results in altered morphology [16]. 

Utilising HTS and a portion of the unique and lead-like NatureBank marine fraction library, we have identified axinellamines A and B as new anti-tubercular natural products. These compounds are highly effective against *M. tuberculosis* in environments with low protein and simple carbon sources, similar to that of the intracellular environment. We observed no apparent cytotoxicity at 25 µM and an almost 90% reduction in intracellular bacterial numbers post-treatment. These data taken together highlight the axinellamine structure class as a promising new anti-TB pharmacophore for future drug development.

## 4. Materials and Methods

### 4.1. General Chemistry Experimental Procedures

Specific rotations were recorded using a JASCO P-2000 polarimeter (Tokyo, Japan). NMR spectra were recorded at 25 °C on a Bruker AVANCE III HD 800 MHz NMR spectrometer (Billerica, MA, USA) equipped with a cryoprobe. The ^1^H and ^13^C chemical shifts were referenced to solvent peaks for DMSO-*d_6_* (δ_H_ 2.50, δ_C_ 39.51). LRESIMS data were recorded on an Ultimate 3000 RS UHPLC (Waltham, MA, USA) coupled to a Thermo Fisher Scientific ISQEC single quadrupole ESI mass spectrometer (Columbia, MA, USA). GRACE Davisil (35–70 µm, 60 Å) C_18_-bonded silica was used for pre-adsorption prior to reversed-phase HPLC. The chromatography resin with pre-adsorbed material was packed into a stainless-steel guard cartridge (10 × 30 mm) and then attached to an HPLC column prior to fractionation. A Waters 600 pump fitted with a Waters 996 photodiode array detector (Milford, MA, USA) fitted with a Gilson 717-plus autosampler (Middleton, WI, USA) was used for RP-HPLC separations. Frozen marine biota was dried using a Dynamic FD12 freeze dryer (Vineyard, NSW, AUS) and ground using a Fritsch Universal Cutting Mill Pulverisette 19 (Pittsboro, NC, USA), or by hand using a granite mortar and pestle. The ground marine biota was extracted at room temperature using an Edwards Instrument Company Bio-line orbital shaker (Narellan, NSW, AUS) set to 200 rpm. Solvents were removed from the crude marine extracts with a Buchi R-144 rotary evaporator (Billerica, MA, USA) and from the HPLC fractions using a GeneVac XL4 centrifugal evaporator (Ipswich, SF, UK). All solvents used for chromatography, MS, and ${\left[\alpha \right]}_{D}$ were from Honeywell Burdick & Jackson or Lab-Scan and were HPLC-grade. H_2_O was filtered using a Sartorius Stedium Arium^®^ Pro VF ultrapure water system (Gottingen, NI, GER).

### 4.2. Marine Sponge Material: Extraction, Fractionation, and Compound Characterisation

The Haplosclerida sponge (NB6020563; AIMS 22282) was collected by trawling at a depth of 60 m off the coast of Broome, Western Australia, Australia, on the 5th of February 1998, by the Australian Institute of Marine Science (AIMS). The sponge sample was immediately frozen at −20 °C upon collection and subsequently transported to the Griffith Institute for Drug Discovery, where the material was freeze-dried and ground into a fine powder and then stored in the NatureBank biota repository. 

The freeze-dried and ground specimen of Haplosclerida (1 g) was sequentially extracted with *n-*hexane (21 mL), CH_2_Cl_2_:MeOH (8:2, 21 mL), and MeOH (39 mL) at room temperature. The CH_2_Cl_2_ and MeOH extracts were combined and dried to give a crude extract (143.1 mg). This extract was pre-adsorbed to C_18_-bonded silica (~1 g) and then packed into a guard cartridge for separation using a C_18_-bonded silica Betasil HPLC column. Isocratic solvent conditions of 90% H_2_O (0.1% TFA)/10% MeOH (0.1% TFA) were initially employed for the first 10 min, followed by a linear gradient to 100% MeOH (0.1% TFA) over 40 min, and a final isocratic condition of 100% MeOH (0.1% TFA) for an additional 10 min at a flow rate of 9 mL/min, collecting at 1 min intervals. In total, 60 fractions were collected.

From each of the 60 fractions, aliquots (250 μge/μL) were transferred into a 96-well microtiter plate and tested in an established HTS assay (see Section 4.4 below for details); fractions 41–43 contained anti-TB activity. Fractions 41 and 43 yielded the TFA salts of the previously described marine natural products [12] axinellamine A (**1**, 3.0 mg, 0.33% dry wt, t_R_ = 41–42 min, purity > 95%) and axinellamine B (**2**, 2.0 mg, 0.22% dry wt, t_R_ = 42–43 min, purity > 95%), respectively.

TFA salt of axinellamine A (**1**): yellow gum; $\;{\left[\alpha \right]}_{D}^{22}\;$ = −22.3° (*c* 0.27, MeOH), lit. $\;{\left[\alpha \right]}_{D}^{20}$ = –18° (*c* 0.16, MeOH) [12]; ^1^H NMR and UHPLC-MS data (Appendix A).

TFA salt of axinellamine B (**2**): yellow gum; $\;{\left[\alpha \right]}_{D}^{22}$ = −26° (*c* 0.26, MeOH), lit. $\;{\left[\alpha \right]}_{D}^{20}$ = –7° (*c* 0.21, MeOH) [12]; ^1^H NMR and UHPLC-MS data (Appendix A).

### 4.3. Bacterial Strains and Culture Conditions

*M. smegmatis* strain MC^2^-155 harbouring the cytoplasmic plasmid pMV261-mCherry was utilised in the initial screening experiments. Virulent *M. tuberculosis* strain H37Rv was utilised in all subsequent screenings and assays. All work with *M. tuberculosis* H37Rv was performed under PC3 laboratory conditions at the Tuberculosis Research Laboratory, University of Queensland. Mycobacterial strains were routinely cultured in Middlebrook 7H9 broth supplemented with 10% (*v*/*v*) ADC (5% bovine serum albumin, 2% dextrose, 0.003% catalase, and 0.85% sodium chloride solution), 0.2% glycerol, and 0.02% (*v*/*v*) Tyloxapol (Sigma-Aldrich, St. Louis, MI, USA), or a defined minimal media (DMM) containing KH_2_PO_4_ (1 g/L), Na_2_HPO_4_ (2.5 g/L), L-asparagine (0.5 g/L), ferric ammonium citrate (5 mg/L), 40 µM MgSO_4_, 4.5 µM CaCl_2_, 0.45 µM ZnSO_4_, 0.1% glycerol, and 0.02% Tyloxapol (Sigma-Aldrich) at 37 °C in standing cultures.

### 4.4. High-Throughput Screening of the NatureBank Fraction Library

The marine invertebrate fraction library was sourced from NatureBank, Griffith Institute for Drug Discovery [11]. Fractions were added to 96-well plates at 250 µge/mL (µg equivalent details described elsewhere [10]). *M. smegmatis* was grown to mid-exponential phase (Optical Density (OD_600_) 0.4). Bacteria were diluted to OD_600_ 0.004 in 7H9s media (7H9 with 10% ADC, 0.5% glycerol, 0.05% Tween-80, and 1% tryptone) and 100 µL added to each well of library plates. Plates were incubated at 37 °C for 48 h. Fluorescence was measured on a FLUOstar Omega plate reader (BMG Labtech, Ortenberg, Germany) with an excitation wavelength of 584 nm and an emission wavelength of 620 nm. Growth inhibition was calculated relative to the positive control wells (no compound) minus the negative control wells (no bacteria).

### 4.5. Bioassays Used to Identify Active HPLC Fractions following Large-Scale Extraction and Isolation Studies on 10 Prioritised Hit Biota Extracts and Chromatographic Fractions

HPLC-generated fractions were added to microtiter plates at 2.5 µge/µL in 100 µL of 7H9s media. *M. tuberculosis* was grown to mid-exponential phase (OD_600_ 0.4 to 0.8). A bacterial single-cell suspension was made by vortexing bacterial cells with ~1 g of glass beads for 90 s, followed by 5 µm filtration. Bacteria were diluted to OD_600_ 0.002 in 7H9s, and 100 µL was added to each well of the microtiter plates. Plates were incubated at 37 °C for 5 days in a humidified incubator before the addition of resazurin solution (30 µL of 0.02% resazurin and 12.5 µL of 20% Tween-80), followed by a further 24 h incubation. Fluorescence was measured on a FLUOstar Omega plate reader (BMG Labtech) with an excitation wavelength of 530 nm and an emission wavelength of 590 nm. Growth inhibition was calculated as in the HTS. 

MIC was determined via the resazurin microtiter plate assay as previously described [19]. The assay was conducted either in 7H9s media or DMM. In the assays where the effect of additional serum was assessed, DMM was supplied with 10% foetal bovine serum (Bovogen). Plates were incubated for 5 days at 37 °C in a humidified incubator before the addition of resazurin solution, followed by a further 24 h incubation before measurement of fluorescence.

### 4.6. Cytotoxic Assay

The murine macrophage cell line RAW267.4 was cultured in Dulbecco’s Modified Eagle’s Medium (DMEM) supplemented with 1 mM sodium pyruvate, 1× GlutaMAX^TM^ (Gibco), and 10% (*v*/*v*) heat-inactivated FBS at 37 °C in a humidified incubator with 5% CO_2_. The cells were passaged as they reached 80% confluence. Macrophages were seeded into microtiter plates and allowed to adhere overnight. Compounds were added to wells at indicated concentrations, with an equivalent vehicle added as a control. Cells were incubated for 24 h before media was removed and replaced with 50 µL 1 mg/mL MTT (3-[4,5-dimethylthiazol-2-yl]-2,5-diphenyltetrazolium bromide) and incubated at 37 °C with 5% CO_2_ for 3 hours. An acidified ethanol solution was added to the assay plate and incubated at room temperature for 15 min with gentle agitation. Absorbance was read at 590 nm, and macrophage survival was calculated relative to the vehicle control.

### 4.7. Intracellular Survival Assay

RAW264.7 macrophages were seeded in microtiter plates at 2.5 × 10^5^ cells/mL in supplemented DMEM and allowed to adhere overnight. A single-cell *M. tuberculosis* suspension was prepared as for the bioassays. Macrophages were infected at a multiplicity of infection of 10 and carried out for three hours. Macrophages were washed three times with phosphate-buffered saline to remove extra-cellular bacteria before the addition of supplemented DMEM with the compounds at the indicated concentrations. Twenty-four hours post-infection, the macrophages were washed three times with phosphate-buffered saline and lysed with 0.1% Triton-X 100. Lysates were serially diluted and plated for enumeration on 7H10 agar (Middlebrook 7H10 supplemented with 10% OADC [0.05% oleic acid, 5% bovine serum albumin, 2% dextrose, 0.003% catalase, and 0.85% sodium chloride solution] and 0.5% glycerol).

## Figures and Tables

**Figure 1 marinedrugs-22-00298-f001:**
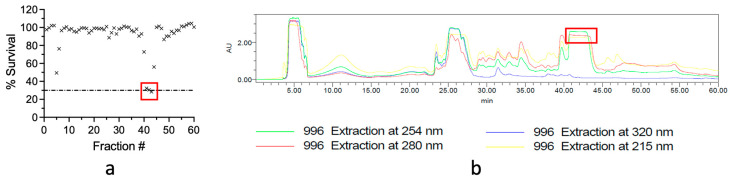
*M. tuberculosis* inhibition by bioassay-guided fractions from NatureBank Haplosclerida extract. (**a**) Inhibition of *M. tuberculosis* growth by fractions from extracts of Haplosclerida. Red box identifies fractions with ≥70% inhibition. (**b**) Reversed-phase HPLC trace of Haplosclerida extract. Red box indicates fractions that were inhibitory to *M. tuberculosis*.

**Figure 2 marinedrugs-22-00298-f002:**
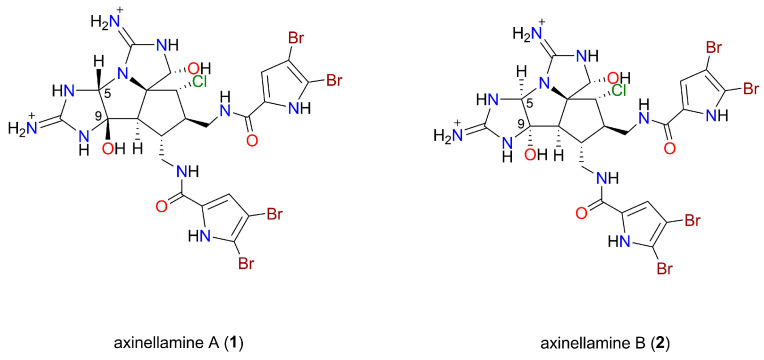
Chemical structures of axinellamines A (**1**) and B (**2**). Both compounds were purified as their TFA salts.

**Figure 3 marinedrugs-22-00298-f003:**
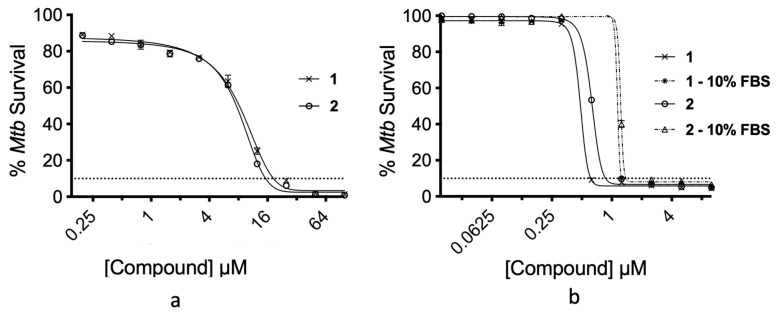
Anti-tubercular activity of axinellamines A (**1**) and B (**2**). (**a**) Inhibition of *M. tuberculosis* growth by **1** and **2** in complex 7H9s growth media. (**b**) Inhibition of *M. tuberculosis* growth by **1** and **2** in minimal growth media ± foetal bovine serum (FBS).

**Figure 4 marinedrugs-22-00298-f004:**
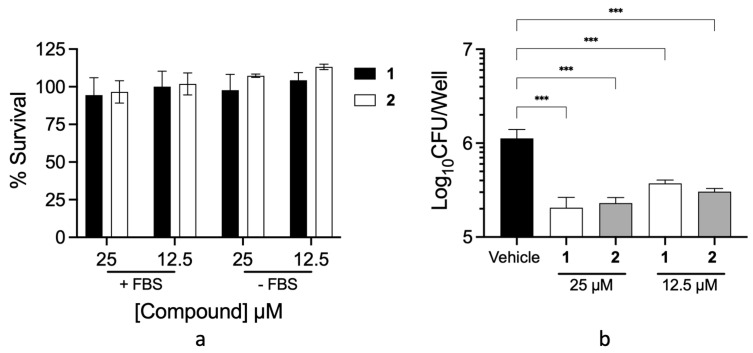
Axinellamines A (**1**) and B (**2**) inhibit *M. tuberculosis* growth in RAW264.7 macrophages. (**a**) MTT assay of RAW264.7 macrophages treated with indicated concentrations of **1** or **2**. (**b**) Intracellular enumeration of *M. tuberculosis* after 24 h treatment with **1** or **2**. *** represents *p* < 0.001 as determined by one-way ANOVA.

**Table 1 marinedrugs-22-00298-t001:** Source of the top 10 fraction hits following HTS of a subset of the NatureBank marine fraction library against *M. smegmatis*.

Phylum	Class	Order	Family	Genus	Species
Cnidaria	Anthozoa	Actinaria			
Porifera	Demospongiae	Verongida	Pseudoceratinidae	*Pseudoceratina*	
Porifera	Demospongiae	Haplosclerida			
Chordata	Ascidiacea	Aplousobranchia	Didemnidae	*Lissoclinum*	*badium*
Porifera	Demospongiae				
Porifera	Demospongiae	Halichondrida	Axinellidae	*Homaxinella*	
Porifera	Demospongiae	Halichondrida	Axinellidae	*Cymbastela*	
Porifera	Demospongiae	Haplosclerida	Callyspongiidae	*Callyspongia*	
Bryozoa					
Porifera	Demospongiae	Poecilosclerida	Microcionidae	*Clathria*	*transiens*

## Data Availability

The raw data supporting the conclusions of this article will be made available by the authors upon request.

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
