# Peer review of "Identification of Axinellamines A and B as Anti-Tubercular Agents"

_marinedrugs, 2024, doi:10.3390/md22070298_

Round 1
Reviewer 1 Report
Comments and Suggestions for Authors
After a high-throughput screening (HTS) of a subset of the NatureBank marine fraction library (2,000 samples), the authors identified a sample with promising anti-mycobacterial activity. This sample was derived from an Australian marine sponge belonging to the order Haplosclerida. Bioassay guided fractionation of the organic extract from the above Haplosclerida sponge resulted in the known, previously identified antimicrobial pyrrole alkaloids, axinellamines A (1) and B (2), which showed a 90% minimum inhibitory concentration (MIC90) of 18 and 15 µM, respectively. After removal of protein and complex carbon sources the MIC90 values seemed to be more advantageous: 0.6 and 0.8 µM, respectively. Axinellamines A and B may be a starting point to identify an anti-tubercular lead compound for future medicinal chemistry efforts.
The manuscript is suitable for publiction after a minor revision.
Line 19: concentration
line 103: a reference for resazurin-reduction assay is missing.
line 121: a reference for MTT assay is missing.
Line 161: „In line with previous reports…” references are missing.
Line 167: Both rifampicin and isoniazid, the current front-line drugs used for TB treatment… The structure of rifampicin and isoniazid are missing.
Line 19/: a space is missing.
Author Response
We thank the reviewer for the rapid and comprehensive review. We have addressed the following issues highlighted, as follows.
Line 19: concentration.
Agree. All concentrations present and line now reads correctly.
Line 103: a reference for resazurin-reduction assay is missing.
Agree. A relevant reference has now been included, i.e. Ang, C.W et al., 2020. (now line 113)
Line 121: a reference for MTT assay is missing.
Agree. A relevant reference has now been included, i.e., Kandale, A. et al, 2021. (now line 131)
Line 161: „In line with previous reports…” references are missing.
Agree. A relevant reference has now been included, i.e. Rodriguez, R.A. et al., 2014. (now line 171)
Line 167: Both rifampicin and isoniazid, the current front-line drugs used for TB treatment… The structure of rifampicin and isoniazid are missing.
Disagree. There is no rationale given as to why the structures of leading anti-TB drugs which have been used clinically in the treatment of TB for nearly 60 years. Providing these structures is not relevant to understanding this study and we do not believe will improve the readers understanding of any of the material. A reference is already given to support the point made in regards to rifampicin and Isoniazid.
Line 19/: a space is missing.
We have checked Line 19 and find no errors.
Reviewer 2 Report
Comments and Suggestions for Authors
The review concerns the article type manuscript entitled Identification of Axinellamines A and B as Anti-Tubercular Agents and submitted to Marine Drugs journal (Manuscript ID: marinedrugs-3064781).
The content of the work is consistent with the title. The Axinellamines A and B contain tetrazatetracyclotetradecane skeleton as the main fragment with two azanylideneimidazolidine functional groups and two 4,5-dibromo- 1H-pyrrole-2-carboxamide substituents. The structural difference between Axinellamines A and B is the stereochemistry in the positions of the H and OH substituents at the C6 and C10, in configuration cis in both cases. The stereochemistry and structure determination based on the reference 12 (doi:10.1021/jo981034g), which gives in Supporting Info 1H NMR spectra of the compounds, but stereochemistry is depicted in form of tables containing 2D NMR data (HMBC, COSY, ROESY).
This is the main remark. The work have a potency to add some new spectral information about these two compounds, which can be valuable in a broad sense. Therefore, I suggest to place in Supplementary Materials at least 13C NMR spectra. Also, information about “chiro-optical data” (line 90) is very scarce. Please, add some information, both in the main text as well as in Supplementary.
Overall, the work is readable and coherent.
Author Response
This is the main remark. The work have a potency to add some new spectral information about these two compounds, which can be valuable in a broad sense. Therefore, I suggest to place in Supplementary Materials at least 13C NMR spectra. Also, information about “chiro-optical data” (line 90) is very scarce. Please, add some information, both in the main text as well as in Supplementary.
Author Response.
The structure elucidation of axinellamines A and B along with their full spectroscopic and spectrometric characterization [1D/2D NMR (1H, 13C, COSY, HSQC, HMBC, ROESY), UV, IR, HRMS, optical rotation] has been previously reported by Urban et al in 1999 (reference 12 in the manuscript). Subsequently, the total synthesis of these compounds, which confirmed the chemical structures proposed during the initial structure elucidation studies, has been reported by O’Malley et al in 2008, Rodriguez et al in 2014 and Ma et al in 2016. We have now included the total synthesis papers of axinellamines A and B in our manuscript, the new text and references. The new text reads:
“Furthermore, several syntheses of axinellamines have been reported in the literature” [line 105]
New references have been added to the manuscript:
“O'Malley, D.P.; Yamaguchi, J.; Young, I.S.; Seiple, I.B.; Baran, P.S. Total Synthesis of (±)-Axinellamines A and B. Angewandte Chemie International Edition 2008, 47, 3581-3583, doi:https://doi.org/10.1002/anie.200801138.”
“Rodriguez, R.A.; Barrios Steed, D.; Kawamata, Y.; Su, S.; Smith, P.A.; Steed, T.C.; Romesberg, F.E.; Baran, P.S. Axinellamines as Broad-Spectrum Antibacterial Agents: Scalable Synthesis and Biology. Journal of the American Chemical Society 2014, 136, 15403-15413, doi:10.1021/ja508632y.”
“Ma, Z.; Wang, X.; Ma, Y.; Chen, C. Asymmetric Synthesis of Axinellamines A and B. Angewandte Chemie International Edition 2016, 55, 4763-4766, doi:https://doi.org/10.1002/anie.201600007.”
We are unsure what “new spectral information” is requested by reviewer 2 as we believe all previously reported spectral data is good quality and adequate.
For the current manuscript on the anti-TB activity of these molecules, in order to confirm that we had purified axinellamines A and B, we compared the proton NMR (same solvent as original paper), specific rotation (same solvent), and MS data only. The comparison of minimal data is standard practice in natural products chemistry, and is critical for dereplication purposes. Proton, specific rotation and MS data comparison of known compounds is part of optimized workflow at GRIDD. Our data was essentially identical to the original data reported. We are confident that we have isolated axinellamines A and B during our current studies and see no need to run carbon NMR spectra or acquire other 2D NMR data.
We are unsure what the reviewer means in relation to “information about “chiro-optical data” (line 90) is very scarce.” We have recorded specific rotation data for both axinellamines A and B in the same solvent to that used in the original report by Urban et al, and these data compares favorably. Thus, we have isolated the same stereoisomer to that reported by Urban et al.
Reviewer 3 Report
Comments and Suggestions for Authors
The authors presented the identification of anti-tubercular marine alkaloids, axinellamines A and B, from the prioritized Haplosclerida sponge. The content of this manuscript is well organized. This manuscript contains content that is of interest to experts in this field as well as non-experts. To make this manuscript even better, please consider the following comments.
1. Lines 228 and 230; Please add the assignments of 1H and 13C-NMR chemical shifts of axinellamines A and B. These data will be useful to readers of this journal.
2. Figure 2; The structures of axinellamines A and B are very similar. Please revise the figure so that readers of this journal can distinguish between the two at a glance.
Author Response
Lines 228 and 230; Please add the assignments of 1H and 13C-NMR chemical shifts of axinellamines A and B. These data will be useful to readers of this journal.
As mentioned above in our responses to Reviewer 2, the full spectroscopic and spectrometric characterization [1D/2D NMR (1H, 13C, COSY, HSQC, HMBC, ROESY), UV, IR, HRMS, optical rotation] has been previously reported by Urban et al in 1999 (reference 12 in the manuscript). Additionally, the total synthesis of these compounds, which confirmed the chemical structures proposed during the initial structure elucidation studies, has been reported by O’Malley et al in 2008, Rodriguez et al in 2014 and Ma et al in 2016. We see no need to assign the proton or carbon NMR chemical shifts again.
Figure 2; The structures of axinellamines A and B are very similar. Please revise the figure so that readers of this journal can distinguish between the two at a glance.
Thank you for this good suggestion. We have updated the Figure 2, which shows the chemical structures of axinellamines A and B, we have added the position numbers to C-5 and C-9, and drawn attention to the only differences in the chemical structures by adding the extra text to the revised manuscript. The new text reads:
“Of note, the only structure differences between axinellamines A and B relate to the configurations at C-5 and C-9 (see Figure 2)”